# Geometrically Constrained and Token-Based Probabilistic Spatial Transformers

Anonymous Full Paper
Submission 42

## Abstract

Fine-grained visual classification (FGVC) remains highly sensitive to geometric variability, where objects appear under arbitrary orientations, scales, and perspective distortions. While equivariant architectures address this issue, they typically require substantial computational resources and restrict the hypothesis space. We revisit Spatial Transformer Networks (STNs) as a canonicalization tool for transformer-based vision pipelines, emphasizing their flexibility, backbone-agnostic nature, and lack of architectural constraints. We propose a probabilistic, component-wise extension that improves robustness. Specifically, we decompose affine transformations into rotation, scaling, and shearing, and regress each component under geometric constraints using a shared localization encoder. To capture uncertainty, we model each component with a Gaussian variational posterior and perform sampling-based canonicalization during inference. A novel component-wise alignment loss leverages augmentation parameters to guide spatial alignment. Experiments on challenging moth classification benchmarks demonstrate that our method consistently improves robustness compared to other STNs.

## 1 Introduction

Fine-grained visual classification (FGVC) is critical for applications such as biodiversity monitoring, where automated species recognition from field images [1] can support large-scale ecological studies. In such settings, objects often appear at arbitrary orientations, scales, and positions, frequently embedded in cluttered natural backgrounds. Unlike coarse object recognition, where category boundaries are wide, FGVC depends on subtle morphological cues. This makes FGVC particularly vulnerable to geometric noise: although a single object can undergo infinitely many spatial transformations (e.g., rotations, viewpoint changes), each transformation produces a distinct pixel-level signal. For a classifier that operates on raw images, these transformed views are effectively different inputs, making it challenging to learn a consistent label function. Broadly, existing strategies for handling geometric variability fall into three paradigms:

**Figure 1.** We propose a pseudo-canonicalization framework based on spatial transformers that can be embedded into the training and inference pipelines of vision transformers. Our canonicalization module $\phi$ is trained in an end-to-end fashion with the pre-trained classifier backbone. The pre-trained frozen tokenizer is used to map the transformed input image to visual tokens, which are used to regress an affine transformation matrix $\hat{\mathbf{T}}$ to rectify the image $\mathbf{x}$. This increases robustness by shielding off spatial noise from the downstream model.

**Pseudo-Invariant Models** The most common approach is to promote invariance through data augmentation. By generating semantically redundant samples, models are encouraged to extract consistent features across transformations [2]. Architectures either learn replicated filters specialized to different transformations [3], which consumes capacity, or fuzzy averages over transformed features [4, 5], which regularizes complexity. While effective in practice, strong augmentation can also hurt generalization by biasing feature dominance [6]. Moreover, this strategy places the full burden on the classifier to simultaneously handle all spatial variations during semantic feature extraction. Modern recognition systems cope with this difficulty through a variety of pre- and post-processing steps, extensive data augmentation, and large-scale training protocols [7].

**Equivariant and Invariant Models** A second line of work encodes invariance directly into the architecture through inductive biases. Classical convolutional neural networks (CNNs) are inherently translation-equivariant [8] but remain sensitive to rotations. To achieve rotation equivariance, Group-CNNs redefine features as functions on the group itself, learning banks of filters parameterized by group elements [9]. For continuous groups, Steerable CNNs [10] generalize this idea by expressing filters as linear combinations of basis filters (e.g., circular

harmonics). Beyond hand-specified groups, equivariances can even be meta-learned from data [11]. While elegant, these methods often trade flexibility for efficiency: restricting the filter space can reduce expressivity, while explicitly representing group dimensions increases memory and computational cost. In practice, invariance is often enforced at inference time by pooling over transformations [12], which further increases overhead.

**Canonicalizers** A third paradigm is canonicalization, where each input is mapped to a standardized representative from its equivalence class under group transformations. In effect, canonicalization collapses the continuous family of transformed images into a single canonical orientation, so that recognition can operate on aligned inputs. This strategy avoids the redundancy of augmentation and the rigidity of equivariant architectures. Canonicalization also has biological precedent: the human visual system is known to store canonical orientations of objects for efficient recognition [13].

Recent work has explored both exact and approximate canonicalizers. Exact methods rely on equivariant scoring functions [14, 15], while pseudo-canonicalizers either learn transformations from finite datasets [16] or compute them at test time [17]. Among these, Spatial Transformer Networks (STNs) [18] are particularly appealing. They provide a simple, differentiable mechanism for learning input-dependent transformations, without introducing architectural constraints or modifying the downstream hypothesis space. Despite this flexibility, STNs have been underutilized in modern transformer-based pipelines and are often dismissed as fragile.

**Contributions** We revisit STNs as a backbone-agnostic canonicalization tool. In fig. 1 the high-level procedure is illustrated. Our contributions are multi-folded:

- We propose a transformer-compatible STN design where the frozen tokenizer provides for the localization and the downstream network.

- Schwöbel et al. [19] treat transformation parameters as latent variables with distributions rather than point estimates. We propose a simplified Gaussian variational family for the posterior over transformation parameters.

- We leveraged the composition framework for STNs proposed by Chen et al. [20] and introduced individual bounds to each regressor to stabilize the prediction.

- Instead of sampling of sampling the entire transformation matrix [19], we sample from multiple spatial component distributions and compose the results to further stabilize the prediction.

- We introduced a novel component-wise alignment loss leveraging the spatial alignment of the training dataset.

We demonstrate improved robustness in FGVC under geometric noise, illustrating both the practical value for ecological monitoring and the broader potential of STNs for canonicalization across visual recognition tasks.

**Outline** In section 2, we first introduce some necessary preliminaries in group theory and the technical implications of it. In section 3, we provide an overview over the current state of the art of STNs. With this foundation, we introduce our framework in section 4. We compare our model to the state of the art on two benchmark datasets in section 5. Finally, we conclude the work in section 6.

# 2 Preliminaries

We provide some theoretical background on the foundational theory underlying the geometry of spatial transformations. We only cover the essentials for this paper and refer the interested reader to Hall [21] and Bronstein et al. [22] for a deep learning perspective on the matter.

Classification on a dataset $\mathcal{X}$ often involve transformations that preserve class identity. For instance, an insect rotated by 90° is still the same insect. Such label-preserving transformations are formalized as *symmetries*, which can be described using group theory.

**The Affine Group** A *group $G$* is a set equipped with an associative binary operation, an identity element, and inverses. Many natural transformations of data can be modeled as groups acting on the input space $\mathcal{X}$. Formally, a (left) *group action* of $G$ on $\mathcal{X}$ is a homomorphism $\rho : G \to \mathrm{GL}_n(\mathbb{R})$ (when $\mathcal{X} \subseteq \mathbb{R}^n$), so that $\rho(g) : \mathcal{X} \to \mathcal{X}$ describes the transformation[1] associated with $g \in G$. The *orbit* of $\mathbf{x} \in \mathcal{X}$ is $G\mathbf{x} = \{\rho(g)\mathbf{x} : g \in G\}$, the set of all symmetry-related versions of $\mathbf{x}$. The set of all orbits forms the *orbit space $\mathcal{X}/G$*, which partitions $\mathcal{X}$ into equivalence classes. The *stabilizer* of $\mathbf{x}$ is $\mathrm{Stab}_G(\mathbf{x}) = \{g \in G : \rho(g)\mathbf{x} = \mathbf{x}\}$, the subgroup of elements leaving $\mathbf{x}$ unchanged.

A central example is the *affine group* $\mathrm{Aff}(n)$, the group of all invertible affine transformations on $\mathbb{R}^n$. We focus on planar transformations in 2D, hence

---

[1]Geometric transformations require resampling onto the discrete pixel grid. For clarity we omit these operators in notation, as they add visual clutter and, when composed repeatedly, introduce cumulative discretization error.

$n = 2$. Representations of this group are matrices of the form

$$\rho(g) = \mathbf{T} := \begin{bmatrix} a & b & t_x \\ c & d & t_y \end{bmatrix} = \begin{bmatrix} \mathbf{M} & \mathbf{t} \end{bmatrix}, \quad (1)$$

with $g \in \mathrm{Aff}(2)$, a linear $2 \times 2$ matrix $\mathbf{M} \in \mathrm{GL}_2(\mathbb{R})$, and a translation vector $\mathbf{t} := [t_x, t_y]^\top \in \mathbb{R}^2$.[2] Translation is a additive group $(\mathbb{R}^2, +)$ acting on $\mathbf{x}$ by $\mathbf{x} \mapsto \mathbf{x} + \mathbf{t}$. $\mathrm{Aff}(2)$ acts by either $\mathbf{x} \mapsto \mathbf{M}\mathbf{x} + \mathbf{t}$ or by $\mathbf{x} \mapsto \mathbf{T}\mathbf{x}$, which means it can be can be decomposed as a semidirect product

$$\mathrm{Aff}(2) \cong \mathbb{R}^2 \rtimes \mathrm{GL}_2(\mathbb{R}). \quad (2)$$

$\mathbf{M}$ can be decomposed by factoring[3] it into

$$\mathbf{M} := \mathbf{R}(\theta)\mathbf{S}(s_x, s_y)\mathbf{H}(h_x, h_y)\mathbf{F}(r_x, r_y) \quad (3)$$

$$\text{with rotation} \quad \mathbf{R}(\theta) := \begin{bmatrix} \cos\theta & -\sin\theta \\ \sin\theta & \cos\theta \end{bmatrix}$$

$$\text{and scaling} \quad \mathbf{S}(s_x, s_y) := \begin{bmatrix} s_x & 0 \\ 0 & s_y \end{bmatrix}$$

$$\text{and shearing} \quad \mathbf{H}(h_x, h_y) := \begin{bmatrix} 1 & h_x \\ h_y & h_x h_y + 1 \end{bmatrix}$$

$$\text{and reflection} \quad \mathbf{F}(r_x, r_y) := \begin{bmatrix} r_x & 0 \\ 0 & r_y \end{bmatrix}.$$

Rotation is expressed by the special orthogonal group $\mathrm{SO}(2) = \{R_\theta \in \mathrm{GL}_2(\mathbb{R}) : R_\theta^\top R_\theta = I, \det(R_\theta) = 1\}$, where $R_\theta$ is a rotation matrix parametrized by an angle $\theta \in [0, 2\pi)$. Scaling is a multiplicative group $(\mathbb{R}^+, \cdot)$ acting via $\mathbf{x} \mapsto s\mathbf{x}$ with scalar $s > 0$. Shearing is a subgroup of $\mathrm{GL}_2(\mathbb{R})$ with $\mathbf{h} \in \mathbb{R}^2$. Reflections comprises the finite set $r_x, r_y \in \{-1, 1\}$ performing sign-flips on $\mathbf{M}$.

**Functions under Group Symmetry** Let $\mathcal{Y}$ be a class space associated with $\mathcal{X}$. A classification function $c : \mathcal{X} \to \mathcal{Y}$ is *G-equivariant* if

$$c(\rho(g)\mathbf{x}) = \rho'(g)c(\mathbf{x}), \quad \forall g \in G, \mathbf{x} \in \mathcal{X}, \quad (4)$$

where $\rho$ and $\rho'$ denote the group actions on $\mathcal{X}$ and $\mathcal{Y}$, respectively. The special case where $\rho'(g)$ is the identity yields *invariance*.

While invariance discards structure by collapsing each orbit to an abstract label, *canonicalization* selects a concrete representative from each orbit. Formally, a *canonicalizer* is a map $\phi : \mathcal{X} \to \rho(G)$ such that $\phi(\rho(g)\mathbf{x})^{-1}\rho(g)\mathbf{x} = \mathbf{x}$ for all $g \in G$, and $\phi(\mathbf{x})\mathbf{x} \in G\mathbf{x}$ lies in the same orbit as $\mathbf{x}$. Thus, $\phi$ provides a consistent "canonical form" for each orbit.

A classifier $c$ parameterizing $p_c(y \mid \mathbf{x})$ with $y \in \mathcal{Y}$ can then be *canonicalized* via

$$p_{c,\phi}(y \mid \mathbf{x}) := p_c(y \mid \phi(\mathbf{x})^{-1}\mathbf{x}) \quad (5)$$

ensuring that predictions are invariant to group transformations while preserving the geometric structure of $\mathcal{X}$. We use $\phi(\cdot) = \hat{\mathbf{T}} = \rho(g) \in \mathrm{Aff}(2)$ to denote predicted transformation matrices.

**Target Subgroups** In this paper we focus on the application of biodiversity monitoring of insects, which uses top-down camera capturings. Here, the most relevant transformations are *rotations*, to account for arbitrary orientation under capture, and *scalings*, to handle natural size variation across individuals. We additionally include *shearings* as they approximate perspective distortions induced by camera angle. Translations are excluded since residual localization errors after detection are typically minor. Reflections are likewise omitted: while some species are approximately reflection-symmetric and the discrete and non-compact nature of reflection groups makes them unsuitable for smooth learned canonicalization.

# 3 Spatial Transformers

Spatial Transformer Networks (STNs) [18] aim to *canonicalize* the input $\mathbf{x}$ by learning transformation parameters $\mathcal{T}$ to compose $\hat{\mathbf{T}}$. Concretely, an STN consists of three parts: a *localization network* $\phi : \mathcal{X} \to \rho(G)$ predicting $\hat{\mathcal{T}}$, a *grid generator* constructing a sampling grid from $\mathcal{T}$, and a *sampler* that resamples $\mathbf{x}$ onto this grid via interpolation. This mechanism is fully differentiable and can in principle be attached to any backbone architecture. As $\phi$ is usually not $\mathrm{Aff}(2)$-equivariant, the map $\mathcal{X} \to \rho(G)$ is only approximate, hence a pseudo-canonicalization. The model must be trained on augmented data so that the localization network learns to undo the applied transformations, and early in training the classifier itself is forced to absorb much of the variability (if its not frozen).

As the canonical form of $\mathbf{x}$ depends on its class, the localization network needs to extract salient (class) features from pixel space. This is not only redundant (as the classifier has to do the same) but also demands high-capacity models. This led to the introduction of latent STNs reftifying feature maps instead raw input images or using shared encoders. While the former may cause too much distortion, the later approach behaves much more stable [23]. In practise, STNs are often limited to subgroups of $\mathrm{Aff}(2)$, such as translation ($\mathbb{R}^2$) [24] or rotation ($\mathrm{SO}(2)$) canonicalization [25].

When working with $\mathrm{Aff}(2)$ standard STNs are prone to instabilities including focus on non-salient

---

[2] In practise, homogeneous coordinate for $\mathbf{T}$, $\mathbf{M}$, and $\mathbf{t}$ are used. We omit them here to reduce visual clutter.

[3] We fix this ordering for consistency, though alternative factorizations are possible. This also holds for the shearing matrix $\mathbf{H}_x(h_x) := \begin{pmatrix} 1 & 0 \\ h_y & 1 \end{pmatrix}\begin{pmatrix} 1 & h_x \\ 0 & 1 \end{pmatrix} = \begin{pmatrix} 1 & h_x \\ h_y & h_x h_y + 1 \end{pmatrix}$, which preserves $\det(\mathbf{H}_x(h_x)) = 1$.

regions and poor performance in noisy scenes [19]. To address these limitations, probabilistic extensions model a variational posterior over transformation parameters and approximate expectations via Monte Carlo sampling [19]. The affine group Aff(2) admits canonical coordinate decompositions [26, 27] that map group actions to simpler operations like cyclic angle translations. This motivates component-wise regression approaches that separately predict rotation, scaling, shearing, and translation parameters before composition, which stabilizes training [20] and allows for parameter domain constraints. STNs can be stacked avoiding information-losing resampling chains [28], and diffeomorphic extensions enable smooth, invertible transformations that prevent collapses to degenerate solutions (e.g. the identity or background alignment) [29]. Further generalizations include pixel-wise motion fields [30] that enable local deformations, though at the cost of increased instability due to the expanded transformation space complexity.

## 4    Methodology

Let $(\mathbf{x}, y) \sim \mathcal{D}$ represent a sampled tuple from a dataset $\mathcal{D}$, where $\mathbf{x} \in \mathbb{R}^{C \times H \times W}$ is an image and $y \in \mathbb{N}$ is its corresponding class label. Here, $C \in \mathbb{N} \setminus \{0\}$ is the number of channels, usually $C = 3$ for RGB images, and $H, W \in \mathbb{N} \setminus \{0\}$ the spatial dimensions of the image. In fig. 2 we provide an overview of our proposed pipeline.

**Shared Tokenizer**    As discussed in section 3, a major downside of STNs is the complex and redundant feature extraction from pixel space to transformation space. Hence, we build on the findings of Finnveden et al. [23] and leverage a shared encoder for both the localization network $\phi$ and the classifer $c$. As the same (latent) token space is used by $\phi$ and $c$ the redundant encoding is avoided to some extent. In this paper, we model $c$ by a vision transformer encoder. We share the frozen tokenizer, which performs standard patchification and linear projection (and positional encoding) of input images. Since our module outputs rectified images that maintain the same spatial structure as the input, downstream transformer models process the transformed data without requiring architectural modifications. This design ensures agnosticism to both the choice of $c$ and the specific downstream task, allowing for flexible deployment across diverse vision applications while preserving the benefits of pre-trained representations.

**Localization Encoder**    We employ a shallow (2-layer) vision transformer encoder to extract high-frequency features and context (like the shape of an

**Figure 2.** We leverage the frozen tokenizer of the backbone to obtain patch tokens, which are processed by a localization encoder into a shared representation. Separate regression heads predict parameters for rotation, scaling, and shearing, which are composed into an affine matrix $\hat{\mathbf{T}}$. The input image is rectified via $\hat{\mathbf{T}}^{-1}$, re-tokenized (with tied weights), and passed to the downstream vision transformer for classification. This design ensures stable, interpretable canonicalization while remaining backbone-agnostic.

object). This performs an automorphism from tokens to tokens $\mathbb{R}^{N \times C} \to \mathbb{R}^{N \times C}$, where $N$ is the number of tokens and $C$ the embedding dimension. We apply a permutation-invariant pooling over $N$ (we pool by averaging) $\mathbb{R}^{N \times C} \to \mathbb{R}^{C}$. This pooled representation is further encoded by a shallow (2-layer) MLP with non-linearities (we use GeLUs). Hence, our shared localization encoder $\phi_{\mathrm{enc}} : \mathbb{R}^{N \times C} \to \mathbb{R}^{C}$ maps tokens (used for the downstream task as well) to high-frequency features to quantify planar pose properties. It provides common feature representations across all spatial regression heads, enabling efficient parameter sharing while maintaining the capacity to capture complex spatial dependencies required for robust affine parameter regression.

**Regression Heads**    We compose $\phi_{\mathrm{enc}}$ with individual regression heads (defined next) to form the localization network $\phi$. As shown in eq. (2) and eq. (3), the affine matrix decomposes into a translation vector, a scaling, shearing, reflection, and

rotation matrix. This decomposition is particularly advantageous in neural networks as it enforces geometric constraints naturally: rotation preserves orthogonality and area, scaling maintains axis alignment while changing area proportionally, shearing preserves area but distorts angles, and each parameter has a clear geometric interpretation, facilitating both optimization and interpretability in spatial transformer networks.

This parametric decomposition allows us to regress each transformation component separately and compose the affine transformation $\mathbf{T}$ through matrix multiplication. We employ separate regression heads for anisotropic scaling $\mathbf{s} = (s_x, s_y) \in \mathbb{R}^2_{++}$, anisotropic shearing $\mathbf{h} = (h_x, h_y) \in \mathbb{R}^2$ and the rotation angle $\theta \in [0, 2\pi)$. Technically, another head for reflection $\mathbf{r} \in \{-1, 1\}^2$ and one for the translation $\mathbf{t} \in \mathbb{R}^2$. As discussed earlier, in this paper we focus on rotation, scaling and shearing. Assuming $\mathbf{s}$ to be regressed in log-space[4], we can form a canonical coordinate frame

$$\mathcal{T} := \{\theta\} \cup \{\log s \in \mathbf{s}\} \cup \{h \in \mathbf{h}\} \qquad (6)$$

of the affine group Aff(2), providing a minimal and interpretable representation.

All heads consist of a linear layer projecting the shared representation $\mathbf{z} \in \mathbb{R}^C$ to a component-specific representation. We add component-specific non-linearities and domain constraints to ensure mathematical validity and numerical stability:

$$\theta = \lambda_\theta \pi \tanh(\mathbf{W}_\theta \mathbf{z} + \mathbf{b}_\theta) \in [-\lambda_\theta \pi, \lambda_\theta \pi],$$

$$\mathbf{s} = 1 - \frac{1}{\lambda_s} \text{sigmoid}(\mathbf{W}_s \mathbf{z} + \mathbf{b}_s) \in \left[1 - \frac{1}{\lambda_s}, 1\right],$$

$$\mathbf{h} = \frac{\pi}{\lambda_h} \tanh(\mathbf{W}_h \mathbf{z} + \mathbf{b}_h) \in \left[-\frac{\pi}{\lambda_h}, \frac{\pi}{\lambda_h}\right], \qquad (7)$$

where $\mathbf{W}_{\{s,h\}} \in \mathbb{R}^{C \times 2}$ and $\mathbf{W}_\theta \in \mathbb{R}^{C \times 1}$ are weight matrices, $\mathbf{b}_{\{s,h\}} \in \mathbb{R}^2$ and $\mathbf{b}_\theta \in \mathbb{R}^1$ are bias vectors and $\lambda_{\{\theta,h\}} \in (0, 1]$ and $\lambda_s \in (0, 1)$ (to ensure the strict positiveness of $\mathbf{s}$) are hyperparameters[5] to bound the co-domains. This is key as $\det(\mathbf{S}(s_x, s_y)) = s_x s_y$, which lets the size of visual areas alter according to the choice of scale, which resampled to a fixed pixel grid causes information loss.[6] Although $\det(\mathbf{H}) = 1$, $\mathbf{H}$ is not orthogonal, it distort angles and shapes by skewing the coordinate system, again leading to information loss after resampling. Similarly, the mappings for translation and reflection can be defined if required.

**Probabilistic Regression** We replace point estimates with variational posteriors, parameterized as Gaussians whose means and variances are predicted by the regression heads. Each regression head outputs both mean $\mu$ and log-variance $\log \sigma^2$ parameters for the corresponding transformation component. We leverage $\mathcal{T}$ in eq. (8) as the distribution means. The log-variances are obtained similarly by extending each head's output. As $\sigma = \exp\left(\frac{1}{2} \log \sigma^2\right)$, we obtain the standard deviations as follows

$$\sigma_{\mathcal{T}} = \exp\left(\frac{1}{2} \alpha_{\mathcal{T}} \tanh(\bar{\mathbf{W}}_{\mathcal{T}} \mathbf{z} + \bar{\mathbf{b}}_{\mathcal{T}}) - \beta_{\mathcal{T}}\right), \quad (8)$$

where $\bar{\mathbf{W}}_{\{s,h\}} \in \mathbb{R}^{C \times 2}$ and $\bar{\mathbf{W}}_\theta \in \mathbb{R}^{C \times 1}$, $\bar{\mathbf{b}}_{\{s,h\}} \in \mathbb{R}^2$ and $\bar{\mathbf{b}}_\theta \in \mathbb{R}^1$. The linear projection by $\alpha_{\mathcal{T}}, \beta_{\mathcal{T}} \in (0, \infty)$ allows the user to bound the log-variance to a reasonable domain.[7] We have $\log \sigma^2 \in [-\alpha - \beta, \alpha - \beta]$ and hence a variance range of $\sigma^2 \in [\exp(-\alpha - \beta), \exp(\alpha - \beta)]$. We draw samples from the variational posterior using the reparameterization trick [31]

$$\mathcal{T} = \mathcal{T} + \sigma_{\mathcal{T}} \odot \epsilon_{\mathcal{T}} \quad \text{with} \quad \epsilon_{\mathcal{T}} \sim \mathcal{N}(\mathbf{0}, \mathbf{I}), \quad (9)$$

where $\odot$ denotes the Hadamard product. This stochastic parameterization enables gradient-based optimization of the variational objective while maintaining the geometric constraints and interpretability of the transformation decomposition.

**Simplification of P-STNs** The Probabilistic STN (P-STN) [19] extends the Gaussian parameterization of transformation parameters by introducing a Gamma hyperprior on precision, which induces a Student-t marginal posterior over transformations. While this hierarchical approach allows adaptive variance and theoretically provides robustness to outliers, its benefits may be limited in practice. Moreover, the hierarchical prior complicates inference: the KL divergence is no longer closed-form, stochastic estimates add gradient variance, and the coupling of variance and mean often slows convergence and risks posterior collapse. These issues increase computational overhead and can destabilize training. In contrast, a Gaussian formulation retains analytic KL terms, efficient reparameterization, and more stable dynamics. Given that the datasets considered here exhibit modest geometric noise without heavy-tailed misalignments, the Gaussian STN is not only sufficient but likely preferable.

**Geometric Alignment Loss** To ensure robust parameter estimation for the affine transformation components, we define geometric alignment measures tailored to each parameter's intrinsic space. For the angular parameters $\theta$ and $\mathbf{h}$ we employ geodesic-inspired angular losses. Specifically, we use

---

[4]Scales compose multiplicatively and thus become additive (as rotation and shearing) in log-space.

[5]For all our experiments we used full-orbit rotations by setting $\lambda_\theta = 1$. We used $\lambda_s = 0.25$ and $\lambda_h = 0.25$ to bound the scaling and shearing to reasonable ranges.

[6]This is why we omitted positive scaling (zooming-out).

[7]We use $\alpha_{\mathcal{T}} = 4$ and $\beta_{\mathcal{T}} = 6$ in our experiments.

cosine-based circular dissimilarity measures

$$\ell(\theta, \hat{\theta}) = 1 - \cos\left(\hat{\theta} - \theta\right), \qquad (10)$$

$$\ell(\mathbf{h}, \hat{\mathbf{h}}) = \sum_{h \in \mathbf{h}} 1 - \cos\left(\hat{h} - h\right). \qquad (11)$$

For the scale parameters, we utilize standard Euclidean distances

$$\ell(\mathbf{s}, \hat{\mathbf{s}}) = \|\mathbf{s} - \hat{\mathbf{s}}\|_2 \qquad (11)$$

With this we define our geometric alignment loss

$$\mathcal{L}_{\text{align}}(\mathcal{T}, \hat{\mathcal{T}}) = \lambda_\theta \ell(\theta, \hat{\theta}) \qquad (12)$$
$$+ \lambda_s \ell(\mathbf{s}, \hat{\mathbf{s}}) + \lambda_h \ell(\mathbf{h}, \hat{\mathbf{h}}),$$

where $\lambda_\mathcal{T} \geq 0$ weighs the impacts.[8]

**Training** During training we minimize the expected loss

$$\mathcal{L} = \mathbb{E}_{\hat{\mathcal{T}} \sim p_\phi(\cdot|\mathbf{x})}\Big[\mathcal{L}_{\text{NLL}}(y, \hat{y}) + \lambda\,\mathcal{L}_{\text{align}}(\mathcal{T}, \hat{\mathcal{T}})$$
$$+ \lambda'\,\mathrm{D}_{\text{KL}}\left[p_\phi(\hat{\mathcal{T}} \mid \mathbf{x}) \,\|\, q(\hat{\mathcal{T}})\right]\Big], \qquad (13)$$

where $\hat{y} = c(\rho(\hat{\mathcal{T}})^{-1}\mathbf{x})$ and $S$ samples are drawn from $p_\phi(\cdot \mid \mathbf{x})$. The Kullback-Leipner divergence (DKL) is defined component-wise such that

$$\mathrm{D}_{\text{KL}}\left[p_\phi(\mathcal{T} \mid \mathbf{x}) \,\|\, q(\mathcal{T})\right] = \sum_{t \in \hat{\mathcal{T}}} \mathrm{D}_{\text{KL}}\left[p_\phi(t \mid \mathbf{x}) \,\|\, q(t)\right].$$
$$(14)$$

This encourages each predicted parameter distribution $p_\phi(\cdot \mid \mathbf{x})$ to remain close to unit Gaussian priors $q(\theta) = q(s) = q(h) = \mathcal{N}(0, 1)$. The loss in eq. (13) generates gradients for both the localization network $\phi$ and the classifier $c$. While $\mathcal{L}_{\text{NLL}}(\cdot, \cdot)$ and $\mathrm{D}_{\text{KL}}[\cdot \mid \cdot]$ produces gradients for both models, $\mathcal{L}_{\text{align}}(\cdot, \cdot)$ only means to update $\phi$.

By optimizing the average loss across sampled transformations, the encoder learns a pseudo-canonicalizer that maps inputs within a bounded affine neighborhood to consistent canonical forms. Formally, for any $g \in G$ within the bounded domain, the canonicalized classifier satisfies

$$p_{c,\phi}(y \mid \mathbf{x}) \approx_\epsilon p_c(y \mid \phi(\mathbf{x})\mathbf{x}), \qquad (15)$$

with $\epsilon$ denoting the residual canonicalization error.

**Inference** During inference, we treat canonicalization as a latent-variable problem over Aff(2). After training, we obtain a posterior $p_\phi(\hat{\mathcal{T}} \mid \mathbf{x})$. The predictive distribution marginalizes over $\hat{\mathcal{T}}$:

$$p_{c,\phi}(y \mid \mathbf{x}) = \int p_c\left(y \mid \rho(\hat{\mathcal{T}})^{-1}\mathbf{x}\right) p_\phi(\hat{\mathcal{T}} \mid \mathbf{x})\mathrm{d}\hat{\mathcal{T}}. \qquad (16)$$

---

[8]We use $\lambda_\theta = 2$, $\lambda_s = \lambda_h = 0.5$ as rotation is the most challenging to estimate correctly.

**Figure 3.** Samples from the EU-Moths [1] and ECU-Moths [32] FGVC datasets.

In practice we approximate this expectation with $S$ Monte Carlo samples,

$$p_\phi(y \mid \mathbf{x}) \approx \frac{1}{S} \sum_{s=1}^{S} p\left(y \mid \rho(\hat{\mathcal{T}}_s)^{-1}\mathbf{x}\right), \qquad (17)$$

where $\hat{\mathcal{T}}_s \sim p_\phi(\hat{\mathcal{T}} \mid \mathbf{x})$. This enforces approximate affine invariance by marginalizing transformation uncertainty during both training and inference.

# 5 Experiments

We train and evaluate multiple baselines across several benchmark datasets using standardized experimental protocols. Training is conducted on NVIDIA A40 GPUs, with complete implementation details publically available in our source code.

**Architectures** All models employ a pre-trained Swin-Base backbone [33] initialized with ImageNet-1k [34] weights and are optimized using AdamW [35] with a cosine learning rate scheduler [36] starting at $5.0 \times 10^{-5}$ for 100 epochs to ensure convergence. We used the standard Swin configurations with $4 \times 4$ patch sizes, 8 attention heads, MLP ratio of 4, relative positional encoding, and GeLU [37] activations. We implement two localization networks: a token-based 2-layer transformer encoder with average pooling and a convolutional network with batch normalization [38], and adaptive pooling.

**Regularization** Images are resized to $224 \times 224$ pixels, standardized using ImageNet statistics, and augmented with horizontal flipping (50%) and occasional Gaussian blurring or sharpness enhancement (10%). We apply label smoothing (0.1), dropout (20% on the final encoder layer), weight decay (0.05), and gradient clipping at norm 1 for regularization.

**Initialization** Transformation matrices are initialized to identity when predicted directly as in [18], while component parameters are zero-initialized with identity composition. Probabilistic variants initialize log-variances at -2, and we employ reflection

**Figure 4.** Top-1 test accuracy decrease on the vanilla and roto-scaled EU-Moth benchmark. These ablation results highlight the performance drops when disabling or replacing parts of our algorithm.

509  padding to prevent boundary artifacts during spa-
510  tial transformations.

511  **Datasets** We focus our experiments on moth
512  FGVC benchmarks (for biodiversity monitoring sys-
513  tems): Ecuador-Moth [32] comprises 1445 samples
514  and 675 classes and EU-Moth [1] has 1650 samples
515  and 200 classes. Samples from both datasets are
516  shown in fig. 3.

517  Let $\mathcal{D}_{\text{test}} \subset \mathcal{D}$ denote the test set. To evaluate
518  the robustness of our canonicalized classifier $p_{c,\phi}(\cdot)$,
519  we introduce two additional test sets. Both rep-
520  resent augmented variants of $\mathcal{D}_{\text{test}}$. We construct
521  the orbit space $\mathcal{D}_{\text{test}}/(\text{SO}(2) \times \mathbb{R}^2)$. This holds all
522  roto-scaled variants of the original images in $\mathcal{D}_{\text{test}}$.
523  In our experiments, we use a finite version of this
524  orbit space using a subgroup of both $\text{SO}(2)$ and $\mathbb{R}^2$
525  of order $n = 16$. We construct a third test set by
526  first augmenting $\mathcal{D}_{\text{test}}$ by random shearing first and
527  then construct the orbit-space from before.[9]

528  As we augment images of moths by $G =$
529  $\text{SO}(2) \times \mathbb{R}^2 (\times \mathbb{R}^2)$, we expect only trivial stabilizers,
530  hence low "symmetry richness" in general. This
531  results in orbit sizes $|G\mathbf{x}| \approx |G|$, $\forall \mathbf{x} \in \mathcal{D}_{\text{test}}$ accord-
532  ing to the Orbit-stabilizer theorem [39]. Hence, we
533  expect a large test-time performance drop of models
534  unable to cope with these transformations.

## 5.1 Ablation Study

536  We study the performance impact of each key com-
537  ponent of our STN through an ablation analysis.
538  We investigated the top-1 test accuracy gap between
539  vanilla and roto-scaled versions of the EU-Moth [1]
540  benchmark. Our results are presented in fig. 4. Sur-
541  prisingly, we found that the KL divergence term in
542  eq. (14) diminishes top-1 test accuracy when used
543  as our primary objective. We experimented with a
544  weighting term [40] and KL annealing [41], but per-
545  formance remained suboptimal, so we disabled the
546  KL term all together. Given the inherently spatially
547  aligned nature of insect datasets [1, 32], our align-
548  ment loss $\mathcal{L}_{\text{align}}$ in eq. (12) consistently improved

---
[9]This limits the test set sizes to a reasonable level as the size increases with the order of the group when constructing orbit spaces.

**Figure 5.** Top-1 test accuracy on the vanilla and augmented EU-Moth benchmark against different number of samples drawn from the posterior.

**Figure 6.** $\phi$ learned to zoom in and align moths to some extend.

549  performance. Most significantly, we found that the
550  decomposed regression approach in eq. (8) (com-
551  pared to direct regression of $\hat{\mathbf{T}}$ as in [18]) yielded
552  the largest performance gain. The probabilistic re-
553  gression of $\mathcal{T}$ in eq. (9), rather than deterministic re-
554  gression, produced the second-largest improvement.
555  We also evaluated the impact of our token-based lo-
556  calization network and found that using a ConvNet
557  operating on raw pixel space (as employed in nearly
558  all related works) decreased performance.

559  The performance drops observed when disabling
560  or replacing components of our algorithm had
561  greater impact on the roto-scaled test set than on
562  the vanilla version. We believe that our setup learns
563  a more robust canonicalization function, and since
564  we evaluate on the orbit space (roto-scaled test set),
565  misclassification compounds exponentially across
566  test images and affine transformation factors.

567  Following the component-wise ablation study, we
568  explored how the number of samples $S$ drawn from
569  $p_\phi(\hat{\mathcal{T}} \mid \mathbf{x})$ affects performance. We maintained con-
570  sistent sample counts during both training and test-
571  ing (allowing for dynamic variation). Results are
572  shown in fig. 5. Performance increases with addi-
573  tional samples, reaching an optimal peak at $S = 8$,
574  which we adopted for subsequent experiments.

## 5.2 FGVC Benchmarking

576  We compared the downstream performance of our
577  model using top-1 test accuracies to various baselines

**Table 1.** Average top-1 test accuracies on Ecuador-Moth [32] and EU-Moth [1]. Highlighted by (†) are the orbit spaces (rotoscale) of the test sets. These are further augmented by random shearing (††). The relative gains compared to the vanilla baseline are added to ease accessability. All methods use a pre-trained Swin-Base Transformer [33] as its classification backbone.

| Method | Ref | EU | EU† | EU†† | ECU | ECU† | ECU†† |
|---|---|---|---|---|---|---|---|
| Vanilla | - | 96.3 | 90.6 | 90.2 | 76.19 | 62.8 | 61.7 |
| Augmented | - | 97.1 (+0.8) | 95.1 (+4.5) | 95.1 (+4.9) | 72.9 (-3.29) | 71.0 (+8.2) | 70.7 (+9.0) |
| STN | [18] | 96.0 (-0.3) | 94.6 (+4.0) | 94.6 (+4.4) | 67.7 (-8.49) | 65.5 (+2.7) | 65.4 (+3.7) |
| Head-STN | [20] | 97.1 (+0.8) | 95.4 (+4.8) | 95.3 (+5.1) | 72.8 (-3.39) | 70.1 (+7.3) | 69.8 (+8.1) |
| Polar | [26] | 96.0 (-0.3) | 94.7 (+4.1) | 94.7 (+4.5) | 69.5 (-6.69) | 67.7 (+4.9) | 67.6 (+5.9) |
| LogPolar | [26] | 94.9 (-1.4) | 93.9 (+3.3) | 93.8 (+3.6) | 67.9 (-8.29) | 65.7 (+2.9) | 65.6 (+3.9) |
| Diffeo-STN | [29] | 95.8 (-0.5) | 94.9 (+4.3) | 94.9 (+4.7) | 68.5 (-7.69) | 67.0 (+4.2) | 66.8 (+5.1) |
| P-STN (Gamma) | [19] | 95.6 (-0.7) | 93.9 (+3.3) | 93.7 (+3.5) | 69.1 (-7.09) | 66.7 (+3.9) | 65.7 (+4.0) |
| P-STN (Gauss) | [19] | 93.8 (-2.5) | 91.3 (+0.7) | 91.3 (+1.1) | 67.7 (-8.49) | 67.2 (+4.4) | 67.3 (+5.6) |
| Ours (Gamma) | - | 96.5 (+0.2) | 95.2 (+4.6) | 95.1 (+4.9) | 72.8 (-3.39) | **71.7 (+8.9)** | **71.6 (+9.9)** |
| Ours (Gauss) | - | **97.6 (+1.3)** | **96.3 (+5.7)** | **96.2 (+6.0)** | **73.4 (-2.79)** | 71.2 (+8.4) | 71.1 (+9.4) |

in table 1. We trained the Swin-Base classification backbone on the vanilla datasets and on a spatially augmented training set (random shearing, scaling, and rotation). Augmented training is one of the core baselines when modelling robust functions against Aff(2)-perturbations [9, 17] as it usually brings high performance gains and is hard to outperform.

Along this baseline, we include the vanilla STN [18] regressing $\hat{\mathbf{T}}$ directly and a version that regresses $\mathcal{T}$ with seperate heads [20] (Head-STN). We include the Diffeomorphic-STN by Detlefsen et al. [29] but without the Continuous Piecewise-Affine velocity fields [42] as this would differ too much from the global affine transformations employed in this paper. Instead we implemented a diffeomorphic transformation using matrix exponential of the skew-symmetric $\hat{\mathbf{T}}$. Another line of approaches was introduced by Esteves et al. [26], where $\mathbf{x}$ is transformed to polar or log-polar space to extract features. Convolution in log-polar space is equivariant to rotations and scaling, hence suitable to regress $\hat{\mathbf{T}}$. We also included P-STNs [19] as already been introduced in section 4. We also studied the impact of Gaussian prior (ours) v.s. the Gamma hyperprior as in [19].

We found that our model outperformed all other baselines on all six benchmarks. While the Gamma prior only showed higher performances on the augmented ECU [32] test set, the Gauss prior outperformed it on the rest of the benchmarks (by a significant margin of around one accuracy point). Interestingly, all modells fail to improve the top-1 performance on the vanilla ECU. This must originate form some bias in the dataset, which we will investigate in future experiments. In fig. 6 we show some samples of the augmented test set and the produced canonical forms by our $\phi$. Note that, STNs are not trained to align samples under human-like expectation (up-right and centred), instead they aim to minimize the NLL together with the classifier.

## 6 Conclusion

We revisited Spatial Transformer Networks in the context of modern transformer-based pipelines, positioning them as flexible canonicalizers that avoid the rigidity of equivariant models. By decomposing affine transformations into components, constraining their regression domains, and introducing a simple probabilistic formulation, we improve the robustness of STN-based canonicalized classifiers. Our component-wise alignment loss further exploits supervision from augmented training data to guide spatial alignment. Empirical results on fine-grained moth classification benchmarks demonstrate consistent gains in robustness to spatial perturbations. Beyond biodiversity monitoring, the framework is broadly applicable to other domains where geometric variability poses challenges, including medical imaging, aerial surveys, and robotics.

**Limitations and Future Work** Our approach inherits fundamental limitations from STNs, which require augmented training data. Since the canonicalization learned, the classifier predominantly learns from noisy canonical forms, leading to suboptimal spatial alignment. Additionally, our component-wise distance loss requires spatially aligned ground truth transformation parameters, limiting applicability to datasets where such information is unavailable. The method is restricted to planar affine transformations and does not extend to complex spatial deformations or three-dimensional variations. Future work should explore incorporating translation and reflection components, extending to diffeomorphic transformations using velocity fields, and developing self-supervised approaches that eliminate the need for explicit transformation labels.

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
