# OpenReview forum: "Geometrically Constrained and Token-Based Probabilistic Spatial Transformers"
_NLDL.org/2026/Conference — Submitted to NLDL 2026_

### Official Review · Reviewer_GK1R · 2025-09-26

**Rating:** 2
**Confidence:** 4
**Final Rating:** 2
**Final Confidence:** 4

**Summary:**

When a dataset has geometric ambiguities, one approach to get rid of the ambiguities is to use canonicalization, for instance mapping rotated images to a canonical upright frame. This paper improves a version of canonicalization from the literature known as spatial transformer networks (STNs), by using ViTs as the neural network components and also by using a probabilistic setting that simplifies prior work. An image is fed into a small ViT to output mean and variance estimates of five parameters used to define an affine transformation. Using this estimated mean and variance, affine transformations are sampled that can be used to warp the original image before feeding it into a final classifier network (also a ViT). Experiments on two datasets of moth classification show favorable results compared to prior spatial transformer networks.

**Strengths:**

The paper tackles a problem of interest to the community by combining prior work with new ideas.

-	Revisiting STNs with modern vision transformers is an exciting idea.
-	The experimental results indicate that the formulation improves on prior STN-based approaches.
-	A proper ablation study is presented in Figure 4, highlighting the contributions of the proposed method components.

**Weaknesses:**

There is a technical weakness which is crucial, so I will discuss it in detail.

- The decomposition of GL(2) chosen in Eq. (3) is redundant, since it used five continuous values, but GL(2) only has four degrees of freedom. I did not find motivation for why this decomposition was chosen. Redundancy is an issue when defining regression targets for the transformation parameters. If a sampled transformation has two sets of parameters that define it, one of those sets of parameters will be regarded as incorrect by the loss function.
- For instance, setting $\theta=0$, $s_x=\sqrt{2}/2$, $s_y=\sqrt{2}$, $h_x=-1$, $h_y=1/2$, the scaling and “shearing” together define a rotation by 45 degrees, which can also be obtained by just using the rotation matrix, unit scaling and no shearing. It is unclear which of these parameter choices should be regressed for a 45 degree rotation.
- More standard decompositions of GL(2), with four values, include (1) from an SVD decomposition: a rotation followed by axis-aligned scaling followed by another rotation, (2) a variation of the SVD: a rotation followed by scaling in two orthogonal directions, (3) from an LDU decomposition: a shearing along x followed by axis-aligned scaling, followed by a shearing along y.

Further weaknesses:

- A baseline of test-time augmentation (especially for the model trained with only data augmentation) should be included as the proposed method of sampling several transformations from the STN and then averaging the predictions from the classifier could be seen as a form of learning a distribution of transformations for test-time augmentation.
- The datasets considered seem very small. Further, the need for canonicalization in these datasets is not clear from Figure 3, especially the ECU dataset where all moths shown are already in the same orientation.

Typos / other comments:

- It is claimed on line 196 that the "shearing" matrices on line 189 form a subgroup, but this is not the case since multiplying two such matrices will not necessarily have a 1 in the top left entry. Further, the transformation matrix shown on line 189 is not typically called a shearing since it does not in general have a single eigenvalue = 1 (which would correspond to the direction of shearing).
- The list of contributions mixes between present and past tense.
- A major selling point in the paper is the “shared tokenizer”. But the tokenizer is just a single linear layer, so it is not clear that this leads to much reduction of “redundant encoding” as put on line 302.
- Double “of sampling” on line 125.
- It is unclear why the top line in Fig 5 dips at 4 samples.
- On line 077 it is claimed that restricting filter space in CNNs can reduce expressivity, however typically the filters are constructed from a steerable basis of the full filter space. I.e. expressivity is not limited given large enough channel dimension. (Cesa & Weiler, General E(2)-equivariant steerable CNNs)
- On line 206 it is implied that canonicalization is more structure preserving than an invariant function, but this cannot be the case since canonicalization leads to an invariant function, i.e. if $c$ denotes the composition of the canonicalizer and the classifier, then $c$ satisfies equation (4).
- Canonicalization is not well-defined for symmetric objects (see for instance (Dym et al, Equivariant Frames and the Impossibility of Continuous Canonicalization)). The paper could mention this and could even use it to motivate the choice of not canonicalizing wrt reflections (since moths are typically mirror symmetric).

**Final Justification:**

Since there has been no rebuttal of the technical weakness I described, my rating remains.

**Justification:**

The justification for recommending rejection of this paper is first and foremost based on the choice to parameterize the affine group with a redundant parameterization. Since this yields non-unique regression targets it is not an intuitive choice and requires a clear motivation (or comparison with more standard parameterizations).

Secondly, there is a missing baseline of test-time-augmentation which should be included as it is similar to but simpler than probabilistic canonicalization.

---

### Official Review · Reviewer_b24N · 2025-10-07
**Extension of STNs with improved robustness**

**Rating:** 2
**Confidence:** 3
**Final Rating:** 2
**Final Confidence:** 5

**Summary:**

The paper revisits the spatial transformer networks with some clever reimagining for modern ViTs. The key innovation is a probabilistic framing of affine parametrisation for more robust equivariance. While the revamping of STN and P-STN is somewhat incremental, the authors introduce some fresh perspectives that could be interesting for further work. However, there are certain issues with how the paper argues and presents its results that leaves this reviewer unsure if it is ready for publication.

**Strengths:**

1. The motivation is grounded well grounded in theory via group actions, and the preliminaries are well constructed for readers without a strong background in mathematics. The exposition of Aff(2) is carefully constructed, and canonicalisation is well motivated and argued. The decomposition into shearing, scaling, and rotation communicates why the idea makes sense.
2. The Gaussian parametrisation provides a nice bridge to probabilistic theory, and adds a clear path to uncertainty estimation.
3. The method and evaluation is strong, experimental protocols are sensible, and include descriptions of very reasonable modeling choices.
4. The empirical results are convincing, albeit on a rather small scale.

**Weaknesses:**

1. Evaluation is limited to small scale datasets, and it is not clear how the method would scale to existing models. As a study under limited data, the work is able to establish itself as a viable approach, but it lacks the empirical depth to argue convincingly for applications outside transfer for specialised domains. However, the paper seems to argue a bit above the level for which the empirical results fall by invoking applications in medical imaging and robotics. This should ideally be addressed in a camera ready version to avoid overselling the work.
2. The proposed alignment loss is limited to cases where labels exists for the affine parameters. This is a relatively clear supervisory signal that may or may not exist in other domains. While the approach during training may seem robust, real world augmentations and symmetries may not be as simple to canonicalise if the full potential of the method is to materialise, particularly given the authors stated ambition. Arguably, this is inherited by the STN, but is also why the method is often not very much applied in practice.
3. Uncertainty estimates seem to be missing entirely. Either the authors didn't consider this, but this is a potential strength of the method, and seems like a missed opportunity.
4. There is a clear drop in performance on one of the two benchmarks; regarding ECU-Moth, the claim by the authors is simply "This must originate form (sic) some bias in the dataset, which we will investigate in future experiments". This is a quite weak argument, and offers nothing to the reader in terms of understanding why or how the method fails in a relatively small scale benchmark.
5. Some presentation takes a little too much space. While the mathematical exposition is nice, it detracts from the results to some extent.
6. This reviewer is slightly worried about the cost of the Monte Carlo approach, which seems left out of the paper. Presumably, this instantiates some iteration over the model, which could prove costly in practice. A clearer exposition of these computational costs would benefit the quality of the work.

**Final Justification:**

While the paper comes across as competent, the current work has a few issues that needs to be addressed for acceptance. In particular, the sample size is very limited, and in one of the two datasets chosen for evaluation by the authors, the model does not show any benefits. The authors simply say this is due to "dataset bias". The authors are encouraged to expand their evaluations and provide a feasible hypothesis for failure cases, which would significantly improve the scientific contribution of the work.

**Justification:**

Despite coming across as a competent and careful piece of work with some novel ideas, the contribution is ultimately too narrow to warrant acceptance in this reviewer's opinion. The evaluation is confined to two small, highly specific datasets, and the method fails to improve one of them. This makes it difficult to assess whether the proposed approach offers benefits, even in small scale benchmarks (total 3000 images over two datasets).

More concerningly, the discussion of the failure case is unconvincing: attributing the drop in accuracy to “dataset bias” without analysis feels speculative and weakens confidence in the authors’ understanding of their own method. It seems reasonable to consider that ECU-Moth --- with 675 classes and only 1445 training samples --- is prone to underdetermination, and that the approach would benefit with more samples to establish canonicalisation. However, this is a methodological weakness of the experimental design; not inherent "bias".

While this paper modernises STNs for ViTs, it does not revive their broader relevance or demonstrate scalability beyond this constrained study. To make the method more relevant, the experimental results should ideally be more conclusive. While this reviewer is typically more interested in the novel contributions with respect to the modeling approach, there are key issues with the study that does not attain the level of rigor required for this conference.

In summary, the work shows technical competence but lacks both empirical breadth and rigor in interpreting its own results. As it stands, this reviewer will not recommend the paper be accepted for the conference.

---

### Official Review · Reviewer_vrwj · 2025-10-08
**The paper is technically strong and well-written but requires additional evaluation metrics, broader validation, efficiency analysis (including overhead quantification), and experiments on clean datasets to meet publication standards.**

**Rating:** 4
**Confidence:** 4
**Final Rating:** 2
**Final Confidence:** 4

**Summary:**

The paper proposes a probabilistic, component-wise Spatial Transformer Network (STN) integrated with vision transformers to improve robustness in fine-grained visual classification (FGVC) under geometric variability such as rotations, scaling, and shearing. Key contributions include:
* Decomposition of affine transformations into rotation, scaling, and shearing with geometric constraints.
* Gaussian variational modeling of transformation parameters for uncertainty-aware canonicalization.
* A novel alignment loss leveraging augmentation parameters.
* Integration with vision transformers using a shared tokenizer for efficiency.
Experiments on moth classification benchmarks (EU-Moth and ECU-Moth) show improved robustness compared to vanilla STNs, Head-STN, and other canonicalization approaches.
The paper is technically sound and methodologically correct within its stated scope. The mathematical formulation, probabilistic modeling, and integration with transformers are well-justified and implemented. Experimental results strongly support the claim of improved robustness under geometric noise.

**Strengths:**

* Clear motivation and relevance: Addresses geometric variability in FGVC, critical for biodiversity monitoring.
* Technical soundness: Rigorous affine decomposition, probabilistic modeling, and integration with transformer pipelines.
* Comprehensive evaluation, within scope: Includes ablation studies and comparisons with strong baselines.
* Practical design: Backbone-agnostic and leverages shared tokenizers for efficiency.
* Reproducible: Provides clear architectural details, hyperparameters, and training protocols, supporting reproducibility.
* Significance: Addresses a critical challenge in fine-grained visual classification: robustness to geometric variability.
* Clarity: Well-structured and clearly written, with logical flow from motivation to methodology and results.

**Weaknesses:**

* Evaluation Metrics: Only Top-1 accuracy is reported. Standard metrics such as precision, recall, and F1-score are missing, limiting interpretability for imbalanced FGVC datasets.
* Dataset Scope: Experiments are restricted to moth datasets, which may not generalize to other FGVC domains.
* Transformation Coverage: Translation and reflection are excluded, reducing applicability to broader scenarios.
* Efficiency Analysis Missing: Overhead of the proposed component is not quantified. Probabilistic sampling (Monte Carlo), multiple regression heads, and KL divergence introduce computational and memory costs, but inference latency and resource usage are not reported.
* Dependence on Augmented Training Data: The alignment loss requires ground truth transformation parameters, limiting applicability to datasets without such metadata.
* Scope and Generalization: The paper does not study how the proposed model performs on classification tasks without affine transformations (beyond reporting vanilla accuracy in Table 1). There is no dedicated analysis of whether the probabilistic STN introduces unnecessary complexity or harms performance on clean datasets. This limits understanding of trade-offs in real-world scenarios where geometric noise is minimal.

**Final Justification:**

Authors need to add more evaluation and details.

**Justification:**

The methodology presented in paper is technically sound and well-grounded in group theory and probabilistic modeling. Integration with vision transformers is practical and backbone-agnostic. Results demonstrate clear gains under geometric perturbations, supported by ablation studies and comparisons with strong baselines.
However, the evaluation scope is narrow (moth datasets only), and standard classification metrics (precision, recall, F1-score) are missing. Computational overhead introduced by probabilistic sampling and multiple regression heads is not quantified. The paper also does not analyze performance on clean datasets without affine transformations, leaving trade-offs unclear. These gaps limit generalizability and practical adoption.
Author's should address missing metrics, quantify overhead, test on clean and diverse datasets, and clarify trade-offs for deployment.

---

### Official Review · Reviewer_k21x · 2025-10-10
**Incremental Probabilistic Spatial Transformer for fine-grained moth classification but experimentally weak**

**Rating:** 2
**Confidence:** 3
**Final Rating:** 2
**Final Confidence:** 4

**Summary:**

This paper presents a supervised method for fine-grained visual classification (FGVC), which is susceptible to geometric variations, focusing on moth imagery under controlled camera captures. Instead of using equivariant methods, this work proposes to use Probabilistic Spatial Transformers as a stronger alternative for pseudo-canonicalization by leveraging pre-trained transformers and using specific heads to predict 2D rotations, scaling, and shearing to compose an affine transformation matrix for rectifying the input. The proposed method’s main contributions are the component-specific constraints to the transformation regression heads and the transformation-specific alignment losses. The paper validates its method using two moth datasets: Ecuador-Moth and EU-Moth, including two test sets, which consist of roto-scaled and shear-roto-scaled variants of the original test set samples. The results show that the proposed method outperforms prior STNs, except the vanilla STN on the vanilla Ecuador-Moth dataset.

**Strengths:**

The proposed method is incremental, simple, and the design choices are thoroughly discussed and w.r.t. prior work.

The paper includes an ablation study on the main components of the proposed method, which is particularly useful for evaluating the contribution of each component of the proposed method.

The proposed method shows promising preliminary findings by outperforming prior STNs on moth FGVC (Table 1), except for the vanilla STN on the vanilla Ecuador-Moth.

**Weaknesses:**

The method relies on the knowledge of the transformation parameters of the data, which limits its applicability to tasks where such information is unavailable (L640-L643).

The method currently handles only 2D affine transformations (rotation, scale, shear), and does not extend to complex spatial deformations or 3D variations (L643-L646). Please discuss potential extensions to the existing method to address such more complex settings.

The main weakness is the experimentation, particularly the narrow experimental scope. Experiments are conducted on just two controlled moth datasets (with 1445 and 1650 samples each) for a single task. With experiments so constrained, the significance of the results and conclusions is weakened, and the generalisability of the method is questionable. The validation of the method for other tasks, such as segmentation, detection, to name a few, and the validation on other datasets, are needed to justify the paper’s claimed conclusions. It is also not clear how robust the method remains with challenging backgrounds (e.g., cluttered), which typically occur in field images for biodiversity monitoring.

Moreover, the scale of the experiments is relatively modest in terms of dataset and model size, without being clear if the method is scalable, and while is backbone agnostic, this is not empirically validated. What is the generalisation of the method to bigger datasets / larger scale backbones / other architectures, and how do these affect the quality of the learned representations?

The results seem to be measured using one seed, while some performance differences, for instance, between the proposed method using Gamma hyperprior vs the Gaussian prior are limited (and with mixed results). If several seeds are used, with reported standard deviations, it would help determine if the differences are significant. Regarding the Gaussian, its results are not convincing to determine whether it is the optimal choice, and with no insight into why the Gaussian has different performance effects when applied to different methods.

It is unclear why all methods underperform the vanilla method on the vanilla Ecuador-Moth dataset. It is argued that this must originate from some dataset bias (L610), yet this needs to be backed up and further elaborated, especially due to the large performance differences and the drop in augmented training.

It is unclear how to choose the specific lambda for each geometric alignment loss. This is important since most probably the inclusion/exclusion of transformations would necessitate tuning to accommodate other tasks (and new transformation-specific losses, which further limit the transferability of the method).

The paper consists of many typos and repetitions, such as the ‘illustracted’ in L111, ‘later’ in L260, ‘of sampling of sampling’ in L125, ‘seperate’ in L587, ‘form’ in L610, among others. The bibliography style requires editing as it is not consistent, particularly in the way the publication year is provided, and the DOI/URL is selected. For instance, citation [25] is missing a venue with incorrect year placement and citation [37] is missing a URL, among others. Figure 4 ‘Ldist’ does not coincide with the ‘Lalign’ defined in Eq. 12.

**Final Justification:**

After reading the other reviewers’ comments, which are similar in many aspects, and given the absence of the authors’ rebuttal, I remain with my initial review of not recommending acceptance of the paper at its current stage. More specifically, although the paper presents a sound, incremental, well-positioned method w.r.t. previous work, its primary weakness is its experimentation, which is limited to only two moth datasets of a total of ~3000 samples and a single task. The generalisation and scalability of the method are yet to be seen, with the discussion of the results being relatively weak, and the method not improving over the vanilla method for one out of the two baseline datasets. Moreover, the method currently requires the knowledge of the transformation parameters of the data, limiting applicability to scenarios where these are not available, and it currently handles only three 2D affine transformations, which further limits broader applicability.

**Justification:**

Although this work addresses an important problem with a valuable effort in the design of the proposed method, it is experimentally weak to back up the incremental technical advances and broader conclusions. Evaluation is limited to FGVC on two controlled moth datasets with simple backgrounds, with some inconsistent conclusions that lack adequate justification. In my humble opinion, this is insufficient to get a clear conclusion to support the robustness and the practical applicability of the proposed method.

---

### Meta-Review · Area_Chair_pCfp · 2025-10-31

**Recommendation:** Reject
**Confidence:** 5

**Metareview:**

This paper proposes an probabilistic Spatial Transformer Network, and reviewers appreciated the technical effort and the exploration of a valuable problem. However, a consensus emerged that the work requires substantial revision to meet the publication standard. The primary concern was the narrow experimental scope, where reviewers felt that validating the method on more diverse and larger-scale datasets would be essential for demonstrating generalisability, particularly since performance gains over baselines were inconsistent in the current study. Further feedback highlighted key opportunities for methodological refinement, including the justification for the chosen parameterisation of the affine group and the practical limitations of requiring ground-truth transformation labels. As no rebuttal was provided to address these points by the authors, the AC agrees with the reviewers assessment and strongly encourages the authors to incorporate this comprehensive feedback to strengthen the work for a future submission.

---

### Decision · Program_Chairs · 2025-11-05

**Decision:**

Reject

**Comment:**

Based on the reviewers and AC comments, the paper cannot be presented at the conference.